# Optimal Thickness of Double-Layer Graphene-Polymer Absorber for 5G High-Frequency Bands

**Alessandro Giuseppe D'Aloia [1,2,*], Marcello D'Amore [1,2] and Maria Sabrina Sarto [1,2]**

[1] Department of Astronautical, Electrical and Energy Engineering, Sapienza University of Rome, 00184 Rome, Italy

[2] Research Center on Nanotechnology Applied to Engineering of Sapienza (CNIS), Sapienza University of Rome, Via Eudossiana 18, 00184 Rome, Italy

[*] Correspondence: alessandrogiuseppe.daloia@uniroma1.it; Tel.: +39-0644-585-806

**Abstract:** A new analytical approach to optimize the thicknesses of a two-layer absorbing structure constituted by a graphene-based composite and a polymer dielectric spacer backed by a metallic layer acting as perfect electric conductor (PEC) is proposed. The lossy sheet is made by an epoxy-based vinyl ester resin filled with graphene nanoplatelets (GNPs) characterized by known frequency spectra of the complex permittivity. The optimal thicknesses are computed at the target frequencies of 26, 28, and 39 GHz in order to obtain a −10 dB bandwidth able to cover the 5G frequency bands between 23.8 and 40 GHz. The resulting absorbing structures, having a total thickness lower than 1 mm, are excited by transverse magnetic (TM) or electric (TE) polarized plane waves and the absorption performances are computed in the 5G high frequency range.

**Keywords:** two-layers absorber; graphene-based composite; oblique incidence; 5G absorption performances

## 1. Introduction

Over the next decade, the fifth generation of wireless technology, indicated as 5G, will play a key-role in forthcoming telecommunications and mobile networks, allowing for the development of smart integrated networks capable of revolutionizing our daily lives [1]. For instance, the data transmission speed will rise sharply, up to 100 times the speed of current wireless data transmission [2], and the latency, i.e., the time it takes to send data from one device to the next via wireless network, will drop drastically, reaching values as low as 1 ms [3].

Such a revolution requires the allocation of new frequency bands, from some gigahertz up to nearly one hundred gigahertz [3]. As a drawback, the risk of unexpected electromagnetic (EM) interference (EMI) increases. Hence, EMI issues must be taken seriously into account and the use of EM absorbers is a promising EMI countermeasure [4].

The design and realization of new absorbers is becoming crucial and large efforts have been made in the development of new high-performance absorbing materials and screens. In fact, new absorbers should conjugate high EM performances with other characteristics, such as lightness, simple structure, flexibility, resistance to corrosion, and cost effectiveness [4–6].

One of the most investigated absorbing panels is the dielectric Salisbury screen, consisting of a lossy layer followed by a PEC-backed dielectric spacer. However, the main limitation is represented by the thickness that, for frequency applications from a few up to 40 GHz, is in the range of centimeters [7]. In this regard, the use of custom-tailored graphene-based composites as lossy layers allows to overcome this bottleneck. Therefore, two-layers absorbing structures are attracting ever-growing interest and the optimal design of microwave absorbers consisting of a graphene-based custom tailored composite

material and a PEC-backed dielectric spacer is nowadays of primary interest. In particular, the selection of EM properties and thicknesses of both layers are key factors for the realization of high-performing absorbing screens with absorption coefficients lower than −10 dB in a wide frequency band within the 5G spectrum allocations. Thus, the design process is attracting growing interest among scientists and technicians, and efficient simulation tools are urgently needed in order to optimize the EM project of absorbers operating also in the 5G spectrum.

Nowadays, the optimum thicknesses and the EM properties of the layers constituting the absorber structures are estimated using genetic algorithms or linear regression analysis techniques [8–11]. However, these methods need complex numerical procedures, often based on simplifying hypotheses. Furthermore, the resulting absorbing structures can be of difficult realization, compromising the cost-effectiveness and the absorber simplicity.

More simple approaches are focused on the preliminary EM investigation of the lossy layer, which should be characterized by an intrinsic impedance close to the free-space one. Moreover, quite often the optimal thicknesses are selected using tentative numerical methods [12–15] or explicit analytical expressions based on simplifying hypotheses [16–19].

This paper proposes a new analytical approach to optimize the thicknesses of the layers constituting a two-layers absorbing structure. This approach is tested on an absorbing structure consisting of a graphene-based composite and a polymer dielectric layer acting as lossy sheet and spacer, respectively.

In particular, the considered lossy sheet is made of an epoxy-based vinyl ester resin filled with different amounts of graphene nanoplatelets (GNPs), i.e., small stacks of graphene layers having lateral dimensions in micrometer range and thicknesses ranging between 2 nm and 10 nm [15].

GNPs have gained most attention as graphene based nanofillers in polymer composites since they are extremely cheap, easily processable, and characterized by outstanding mechanical properties, along with high thermal and electrical conductivities [16]. Furthermore, they are suitable for large scale production [20] and represent an excellent alternative to the use of pure graphene, which is not yet produced on a large scale and its price is still high [20–22]. The current and future applications of GNPs filled composites are unlimited, from materials with enhanced mechanical and thermal behavior up to new functional materials or conductive coatings, such as sensors, new electronic devices, energy harvesting, and EM shields [6,15,20–23]. Moreover, they allow the production of innovative materials with outstanding electromagnetic properties, suitable for the realization of next generation microwave absorbers [24–26]. Hence, GNPs filled composites are considered in this study. In particular, the new design process is applied to evaluate the optimum configuration of the selected two-layers absorbing structure in the 5G high frequency range up to 40 GHz and to compute absorption performances considering graphene based lossy layers employing different amounts of GNPs.

## 2. Structure of the Graphene-Polymer Absorber

The considered Salisbury screen structure is composed of a graphene-based lossy sheet and a polymer dielectric spacer, followed by a metallic layer acting as a perfect electric conductor (PEC). The polymer spacer and the composite are indicated in Figure 1 as layer 1 and layer 2, respectively.

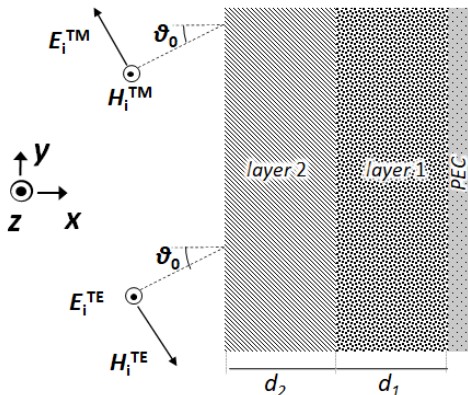

**Figure 1.** Absorber structure consisting of a PEC-backed polymer dielectric spacer (layer 1) and a graphene-based lossy sheet (layer 2). $E_i^{TM(TE)}$ and $H_i^{TM(TE)}$ are the incident electric and magnetic fields in case of TM or TE polarizations.

The absorber is illuminated by a transverse magnetic (TM) or transverse electric (TE) polarized plane wave with incidence angle $\theta_0$, as shown in Figure 1. The absorber is supposed to be of infinite dimensions along the *y*- and *z*- axes.

The spacer layer is made of polypropylene (PP), having relative permittivity $\varepsilon_1 = 2.36$, assumed constant in the considered frequency bandwidth, and relative permeability $\mu_1 = 1$.

The graphene-based lossy sheet, which does not show any magnetic property (hence $\mu_2 = 1$), is constituted by an epoxy-based vinyl ester resin filled with different amounts of GNPs.

The relative permittivity $\varepsilon_2 = \varepsilon_2' - j\varepsilon_2''$ of the resulting GNP composites has been experimentally measured, modelled, and predicted by means of an analytical procedure based on the Maxwell Garnet formula [15,26].

The real and imaginary parts, $\varepsilon_2'$ and $\varepsilon_2''$, of the permittivity are represented in Figure 2a,b, respectively, for composites filled with a GNP weight concentration ranging between 0.25% and 4%.

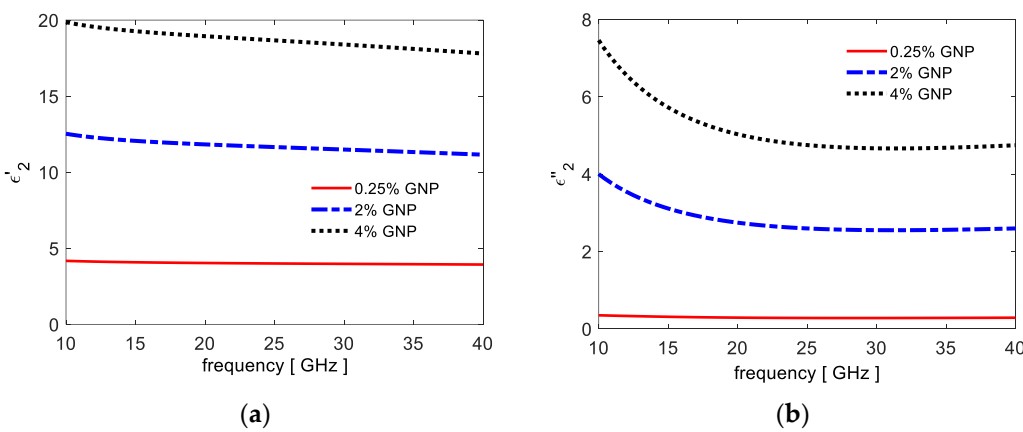

(a)                    (b)

**Figure 2.** Frequency spectra of the real (**a**) and imaginary (**b**) parts of the permittivity of the graphene based composite employing different amounts of GNPs.

## 3. Optimal Absorber Design

The reflection coefficient in case of TM(TE) polarization, $R^{TM(TE)}$, is evaluated as:

$$R^{TM(TE)} = \left| \frac{Z_{in}^{TM(TE)} - \eta_0^{TM(TE)}}{Z_{in}^{TM(TE)} + \eta_0^{TM(TE)}} \right| \tag{1}$$

in which $\eta_0^{TM}$ and $\eta_0^{TE}$ are the TM and TE wave impedances, given by:

$$\eta_0^{TM} \ = \ \eta_0 \, \cos\vartheta_0 \tag{2}$$

$$\eta_0^{TE} \ = \ \frac{\eta_0}{\cos\vartheta_0} \tag{3}$$

being $\eta_0$ the free space wave impedance [25].

The TM(TE) absorber input impedance $Z_{in}^{TM(TE)}$ is computed as:

$$Z_{in}^{TM(TE)} \ = \ \eta_2^{TM(TE)} \frac{Z_{in1}^{TM(TE)} + \eta_2^{TM(TE)}\tanh(\gamma_2 d_2)}{\eta_2^{TM(TE)} + Z_{in1}^{TM(TE)}\tanh(\gamma_2 d_2)} \tag{4}$$

where $Z_{in1}^{TM(TE)}$ is the TM(TE) input impedance of the dielectric spacer layer:

$$Z_{in1}^{TM(TE)} \ = \ \eta_1^{TM(TE)}\tanh(\gamma_1 d_1) \tag{5}$$

The TM and TE wave impedances of the *i*-th layer, with *i* equal to 1 or 2, are computed as:

$$\eta_i^{TM} \ = \ \eta_0 \sqrt{\frac{\mu_i}{\varepsilon_i}} \, \cos\vartheta_i \tag{6}$$

$$\eta_i^{TE} \ = \ \eta_0 \sqrt{\frac{\mu_i}{\varepsilon_i}} \, \frac{1}{\cos\vartheta_i} \tag{7}$$

where $\cos\vartheta_1$ and $\cos\vartheta_2$ are derived from $\sin\vartheta_1$ and $\sin\vartheta_2$ evaluated by means of the Snell's law as:

$$\sin\vartheta_1 \ = \ \frac{\gamma_2}{\gamma_1} \, \sin\vartheta_2 \tag{8}$$

$$\sin\vartheta_2 \ = \ \frac{\gamma_0}{\gamma_2} \, \sin\vartheta_0 \tag{9}$$

in which the propagation constants $\gamma_0$, $\gamma_1$ and $\gamma_2$ are given by:

$$\gamma_0 \ = \ j\frac{\omega}{c_0} \tag{10}$$

$$\gamma_1 \ = \ j\frac{\omega}{c_0}\sqrt{\mu_1\varepsilon_1} \tag{11}$$

$$\gamma_2 \ = \ j\frac{\omega}{c_0}\sqrt{\mu_2\varepsilon_2} \tag{12}$$

In order to minimize the reflection coefficient, the hyperbolic inner impedance at the front surface of the reflector layer should be close to the free space wave impedance. Therefore, the following matching condition should be satisfied:

$$Z_{in}^{TM(TE)} \ = \ \eta_0^{TM(TE)} \tag{13}$$

This equation cannot be resolved analytically, and several approaches were developed to minimize $R$ of single- or multi-layer Salisbury screens.

Indeed, the design process is focused on the analytical expressions of the optimal thicknesses $d_1^*$ and $d_2^*$ for assumed values of $d_2$ and $d_1$ respectively, frequency $f^*$ and incidence angle $\vartheta_0$.

According to the matching Equation (13), the real- and imaginary- part of $Z_{in}^{TM(TE)}$ should be equal to $\eta_0^{TM(TE)}$ and zero, respectively. The obtained transcendental equations cannot be resolved literally to obtain the optimal thicknesses $d_1^*$ and $d_2^*$. Thus, a simplified approach is needed.

In particular, the Equation (4) of $Z_{in}^{TM(TE)}$ and the matching Equation (13) are combined together and then resolved in terms of $d_1^*$ and $d_2^*$ for given $d_2$ and $d_2$ values.

After some developments, the following expressions are obtained:

$$\tanh\left(\gamma_1\cos\vartheta_1 d_1^{TM(TE)}\right) = \Phi_1^{TM(TE)} \tag{14}$$

$$\tanh\left(\gamma_2\cos\vartheta_2 d_2^{TM(TE)}\right) = \Phi_2^{TM(TE)} \tag{15}$$

in which:

$$\Phi_1^{TM(TE)} = \frac{\sqrt{\frac{\mu_2}{\varepsilon_2}}\Gamma_2^{TM(TE)}\tanh(\gamma_2\cos\vartheta_2 d_2) - 1}{\sqrt{\frac{\mu_1\varepsilon_2}{\mu_2\varepsilon_1}}\Gamma_{12}^{TM(TE)}\tanh(\gamma_2\cos\vartheta_2 d_2) - \sqrt{\frac{\mu_1}{\varepsilon_1}}\Gamma_1^{TM(TE)}} \tag{16}$$

$$\Phi_2^{TM(TE)} = \frac{\sqrt{\frac{\mu_1}{\varepsilon_1}}\Gamma_1^{TM(TE)}\tanh(\gamma_1\cos\vartheta_1 d_1) - 1}{\sqrt{\frac{\mu_1\varepsilon_2}{\mu_2\varepsilon_1}}\Gamma_{12}^{TM(TE)}\tanh(\gamma_1\cos\vartheta_1 d_1) - \sqrt{\frac{\mu_2}{\varepsilon_2}}\Gamma_2^{TM(TE)}} \tag{17}$$

with:

$$\Gamma_1^{TM} = (\Gamma_1^{TE})^{-1} = \cos\vartheta_1/\cos\vartheta_0 \tag{18}$$

$$\Gamma_2^{TM} = (\Gamma_2^{TE})^{-1} = \cos\vartheta_2/\cos\vartheta_0 \tag{19}$$

$$\Gamma_{12}^{TM} = (\Gamma_{12}^{TE})^{-1} = \cos\vartheta_1/\cos\vartheta_2 \tag{20}$$

Finally, applying the $(\tanh^{-1})$-function to Equation (14) and to Equation (15), the following analytical expressions of the optimal thicknesses are derived:

$$d_1^{*TM(TE)} = \left|\frac{1}{2\gamma_1}\ln\left(\frac{1+\Phi_1^{TM(TE)}}{1-\Phi_1^{TM(TE)}}\right)\right| \tag{21}$$

$$d_2^{*TM(TE)} = \left|\frac{1}{2\gamma_2}\ln\left(\frac{1+\Phi_2^{TM(TE)}}{1-\Phi_2^{TM(TE)}}\right)\right| \tag{22}$$

In case of normal incidence, the optimal thicknesses $d_i^* = d_i^{*TM} = d_i^{*TE}$, with $i = 1, 2$, assumes the following simple expression:

$$d_i^* = \left|\frac{1}{2\gamma_i}\ln\left(\frac{1+\Phi_i}{1-\Phi_i}\right)\right| \tag{23}$$

in which $\Phi_i$ is given by Equations (10a,b) assuming $\Gamma_i = \Gamma_i^{TM} = \Gamma_i^{TE} = 1$.

It is worth noting that, due to the analytical definition of $(\tanh^{-1})$-function and to the periodicity of $(\tanh)$-function, Equations (21), (22) and (23) permit to evaluate the optimal thicknesses $d_1^*(d_2)$ and $d_2^*(d_1)$ giving rise to the relative minimum value of the reflection coefficient.

## 4. Absorber Design and Performances

The high-frequency bands of 5G next generation technology of UK/Europe/China, Korea and USA are reported in Table 1. For each frequency band, the central frequency $f_c$ is also indicated.

**Table 1.** 5G high frequency bands of some countries.

| Frequency Band [GHz] | $f_c$ [GHz] | Country |
|---|---|---|
| 24.25–27.50 | 25.87 | Europe/UK/China |
| 26.50–29.50 | 28.00 | Korea |
| 27.50–28.35 | 27.92 | USA |
| 37.00–40.00 | 38.50 | USA |

The optimal design is carried out assuming the target frequencies equal to 26 GHz, 28 GHz, and 39 GHz, which are close to the central values $f_c$ reported in Table 1.

Moreover, two different graphene-based composites are selected for the realization of the lossy layers. The first one has a GNP concentration equal to 2% wt. with respect to the total resin amount, the second one employs 4% wt. of GNPs. It is worthwhile to notice that higher GNP amounts lead to technical difficulties, compromising the cost effectiveness and the quality of the resulting layer.

### 4.1. Composite filled with 2% of GNPs.

The optimal thicknesses $d_1^*$ and $d_2^*$ are computed as functions of the given thicknesses $d_2$ and $d_1$, respectively, using Equation (23) for the composite filled with 2% wt. of GNPs in the case of normal incidence.

The obtained results are represented in Figure 3.

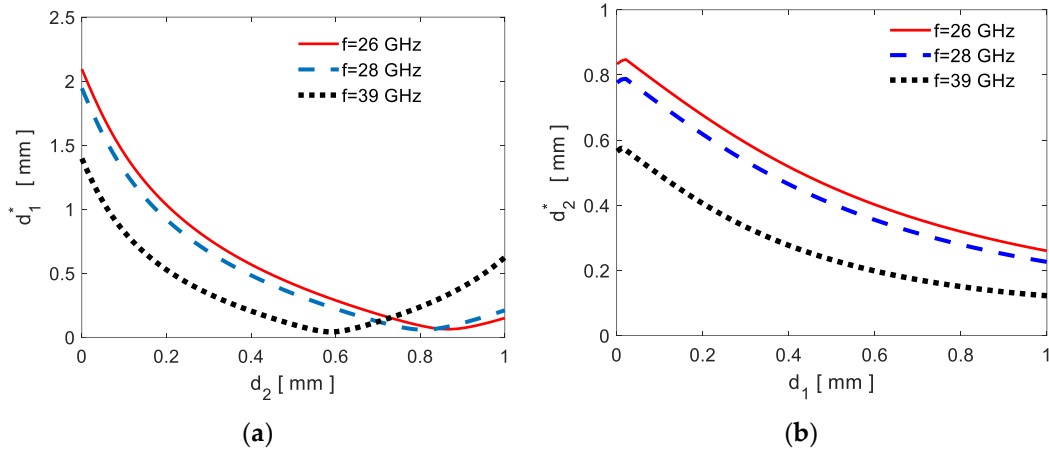

(**a**)    (**b**)

**Figure 3.** Optimal thickness $d_1^*$ as a function of $d_2$ (**a**) and optimal thickness $d_2^*$ as a function of $d_1$ (**b**), computed at the target frequencies of 26 GHz, 28 GHz and 39 GHz.

It should be noted that the same value of $d_1^*$ can be obtained considering two different values of $d_2$ in Figure 3a. The same discrepancy appears in Figure 3b for $d_1$ values lower than 0.5 mm. It yields that Equation (23) is not a one-to-one function, meaning that more thickness combinations can satisfy the matching condition.

In order to clarify and validate the proposed analytical approach, the reflection coefficient $R$ is computed numerically by means of Equation (1) considering all the possible $(d_1, d_2)$ combinations at the target frequencies. As an example, a 3D-map reporting the computed $R$ values for $d_1$ ranging between 0 and 2.5 mm and $d_2$ between 0 and 1 mm at a frequency of 39 GHz is shown in Figure 4a. Successively, the $d_1^*$ value giving rise to the minimum reflection coefficient for each selected $d_2$ value is computed numerically. The obtained curve $d_1^*(d_2)$ (red line) is reported in Figure 4b in which it also appears the curve computed by means of Equation (23) (black dotted line). It is worth noting that the curves are almost overlapping for $d_2$ values lower than 0.6 mm and the relative percent difference between them is always lower than 5% in the decreasing part of the $d_1^*(d_2)$ curve.

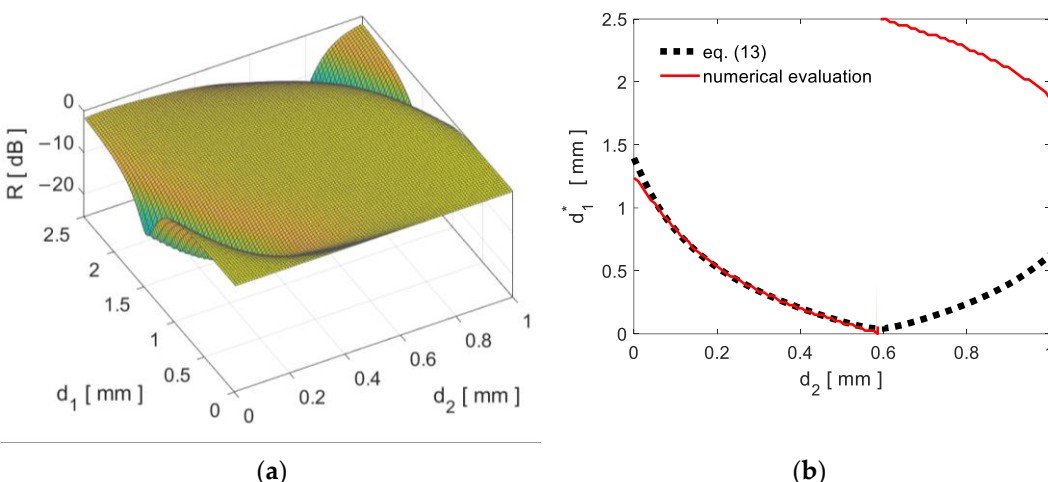

(**a**)　　　　　　　　　　　　　　　　　　　　　(**b**)

**Figure 4.** 3D−map reporting the computed *R* values for all the ($d_1$,$d_2$) combinations in the selected $d_1$ and $d_2$ intervals at the target frequency of 39 GHz (**a**) and optimal thickness $d_1^*$ computed numerically (red line) or by Equation (13) (black dotted line) (**b**).

Figures 5–7 show the $d_1^*$ and $d_2^*$ values as functions of $d_2$ and $d_1$, respectively (blue dashed curves), while the red curves report the corresponding minimum *R* values for the assumed target frequencies.

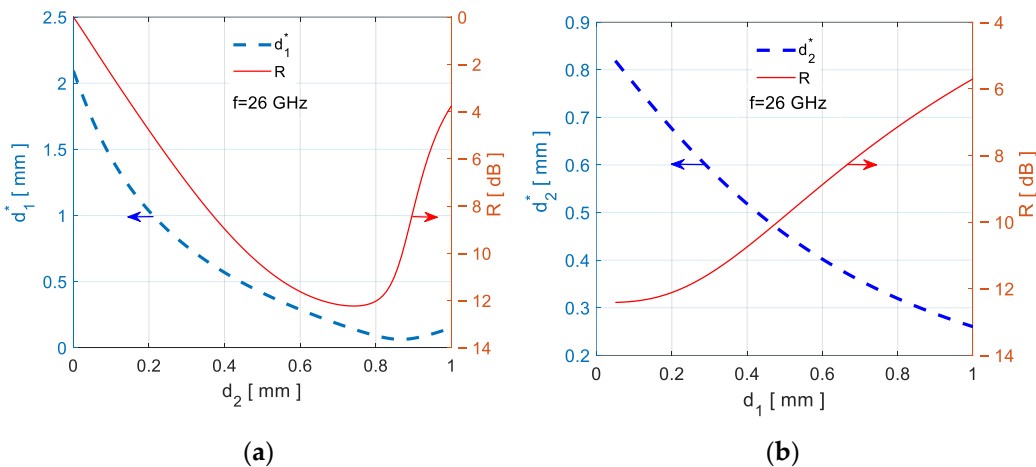

(**a**)　　　　　　　　　　　　　　　　　　　　　(**b**)

**Figure 5.** Optimal thicknesses $d_1^*(d_2)$ (**a**) and $d_2^*(d_1)$ (**b**) at target frequency of 26 GHz.

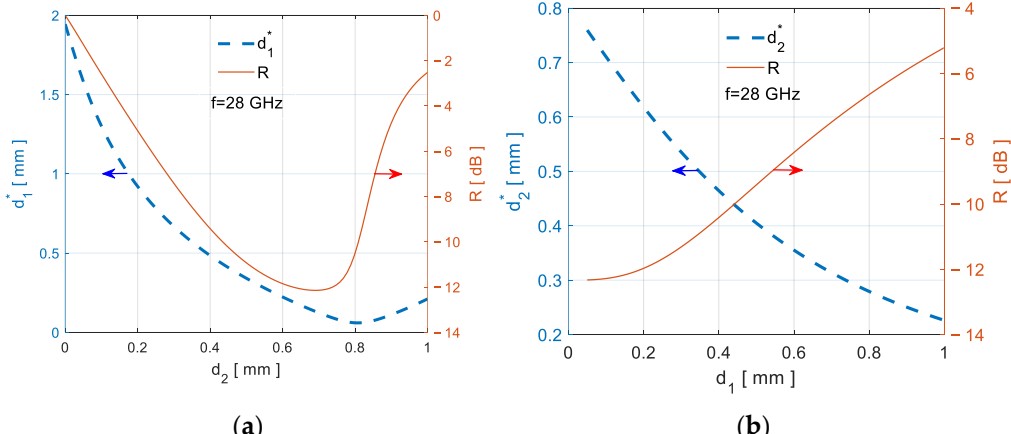

(**a**)　　　　　　　　　　　　　　　　　　　　　(**b**)

**Figure 6.** Optimal thickness $d_1^*(d_2)$ (**a**) and $d_2^*(d_1)$ (**b**) at target frequency 28 GHz.

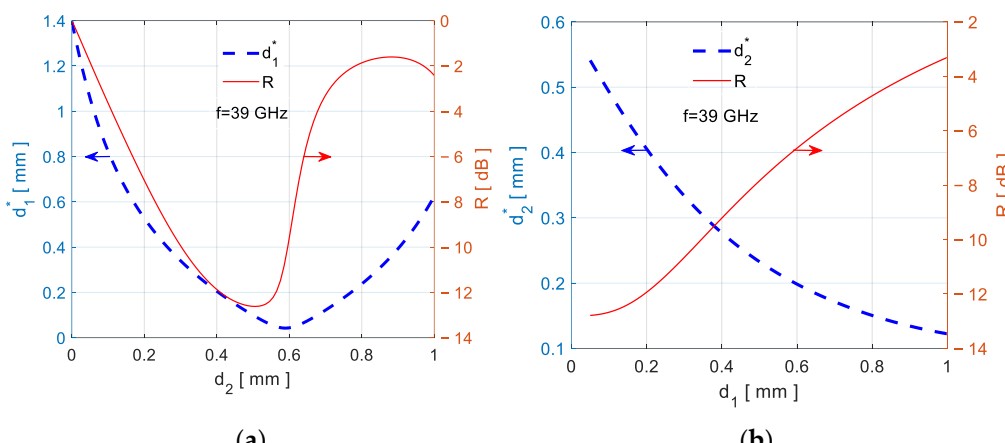

| (**a**) | (**b**) |
|---|---|

**Figure 7.** Optimal thicknesses $d_1^*(d_2)$ (**a**) and $d_2^*(d_1)$ (**b**) at target frequency of 39 GHz.

For instance, according to Figure 5a, obtained at the target frequency $f$ = 26 GHz, assuming $d_2$, i.e., the thickness of the composite sheet, equal to 0.43 mm, the corresponding optimum spacer thickness $d_1^*$ = 0.5 mm should be selected in order to obtain the minimum reflection coefficient $R$ = −11.2 dB. Therefore, the lower minimum value $R^*$ = −12.1 dB is obtained assuming $d_2$ = 0.74 and $d_1^*$ = 0.14 mm. Similarly, from Figure 6a, obtained for $f$ = 28 GHz, and Figure 7a, computed at $f$ = 39 GHz, the lower values $R^*$ = −12.1 dB and $R^*$ = −12.2 dB can be achieved for $d_1^*$ = 0.12 mm and $d_2$ = 0.70 mm or $d_1^*$ = 0.09 mm and $d_2$ = 0.5 mm.

As expected, the same thickness values result from Figures 5b, 6b, and 7b. Notice that when the optimal thickness values are selected in the increasing part of the curves representing $d_1^*$ versus $d_2$ the reflection coefficient increases, resulting in a lower absorption performance. This fact confirms that only the decreasing part of the $d_1^*$ versus $d_2$ curves should be considered.

Table 2 summarizes the obtained values of the lower minimum reflection coefficient $R^*$ corresponding to the selected two-layers thicknesses at the target frequencies. These values are retrieved from Figures 5–7. In the same table, the corresponding bandwidths $\Delta f$ at −10 dB are reported.

**Table 2.** Optimal thicknesses, lower minimum reflection coefficient and bandwidths at −10 dB for composite filled with 2% of GNPs at target frequencies in case of normal incidence.

| Case | $f^*$ [GHz] | $R^*$ [dB] | $d_1^*$ [mm] | $d_2^*$ [mm] | $\Delta f$ [GHz] |
|---|---|---|---|---|---|
| A | 26 | −12.1 | 0.14 | 0.74 | 24.0–27.6 |
| B | 28 | −12.1 | 0.12 | 0.70 | 25.5–29.7 |
| C | 39 | −12.2 | 0.09 | 0.50 | 36.2–43.0 |

It should be noted that the optimal thicknesses $d_2^*$ of the composites reported in Table 2 may be of difficult realization or they may compromise the final absorbing structure flexibility. For this reason, different thicknesses may be preferred with consequent lower absorbing performances.

The real and imaginary parts of the two-layers input impedance given by Equation (3) should be theoretically close to $\eta_0$ and zero, respectively, according to the matching condition of Equation (8). This condition would be more easily satisfied in the case of pure conductive lossy sheets and dielectric spacers.

It should be noted that the GNP-filled composites considered in this work are characterized by conductive and dielectric properties, represented by the real and imaginary parts of the complex permittivity, as shown in Figure 2. As a consequence, the lossy sheet is not pure conductive, and the theoretical matching condition may not be fully satisfied. For instance, at the frequency of 26 GHz and assuming $d_1^* = 0.14$ mm and $d_2^* = 0.74$ mm it results $\text{Real}(Z_{in}) = 546\ \Omega$ and $\text{Imag}(Z_{in}) = -154\ \Omega$, as shown in Figure 8a,b.

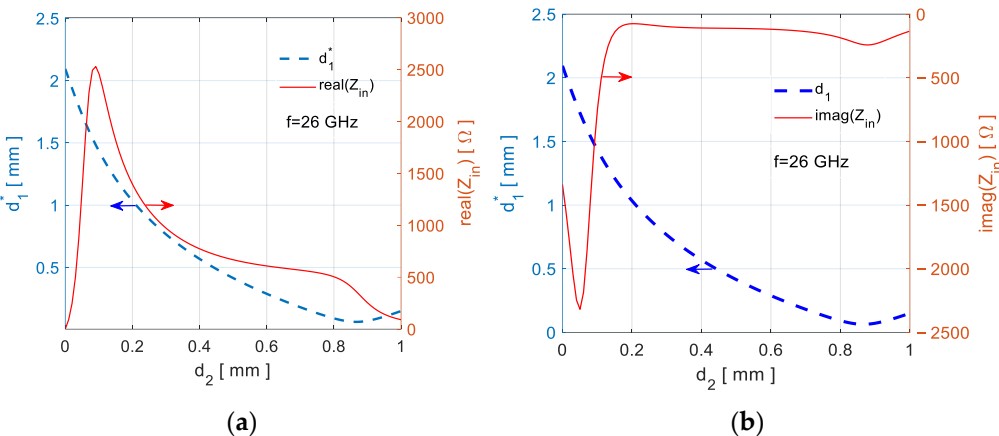

    (**a**)                                (**b**)

**Figure 8.** Real (**a**) and imaginary (**b**) parts of input impedance as function of $d_1^*(d_2)$ at a frequency of 26 GHz.

Finally, the reflection coefficient is computed in the 5G frequency band between 10 GHz and 40 GHz assuming the optimal thicknesses in Table 2. The obtained frequency spectra are represented in Figure 9. The minimum reflection coefficient $R^*$ is obtained at the target frequencies of 26, 28, and 29 GHz, fulfilling the objectives of the optimal design.

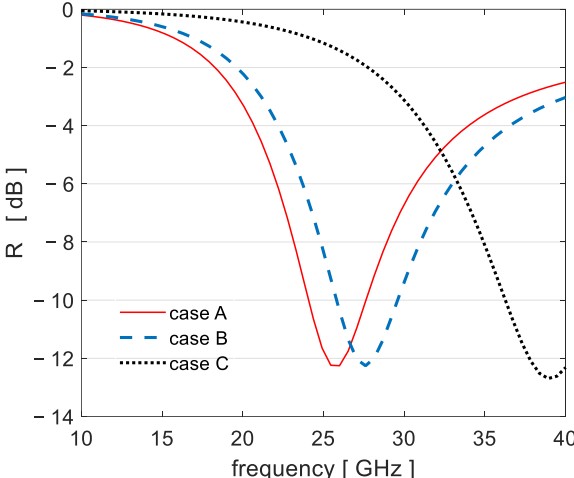

**Figure 9.** Frequency spectra of the reflection coefficient for the cases A, B and C shown in Table 2 for normal incidence.

## 4.2 *Composite filled with 4% of GNPs.*

In order to improve the absorbing performance, the composite filled with 4% of GNPs is considered. Table 3 summarizes the obtained values of the lower minimum reflection coefficients $R^*$ in case of normal incidence at the selected frequencies $f^*$, the optimum thicknesses $d_1^*$ and $d_2^*$ obtained following the procedure described in Section 3 and the corresponding bandwidths $\Delta f$ at $-10$ dB.

**Table 3.** Optimal thicknesses, lower minimum reflection coefficient and bandwidths at −10 dB for composite filled with 4% of GNPs at target frequencies in case of normal incidence.

| Case | $f^*$ [GHz] | $R^*$ [dB] | $d_1^*$ [mm] | $d_2^*$ [mm] | $\Delta f$ [GHz] |
|------|------|------|------|------|------|
| A | 26 | −23.4 | 0.05 | 0.62 | 23.8–28.7 |
| B | 28 | −23.0 | 0.05 | 0.58 | 25.4–30.8 |
| C | 39 | −24.9 | 0.05 | 0.40 | 35.7–42.0 |

　　　The obtained frequency spectra of the reflection coefficient for the cases A, B, and C in Table 3 are represented in Figure 10 for normal incidence.

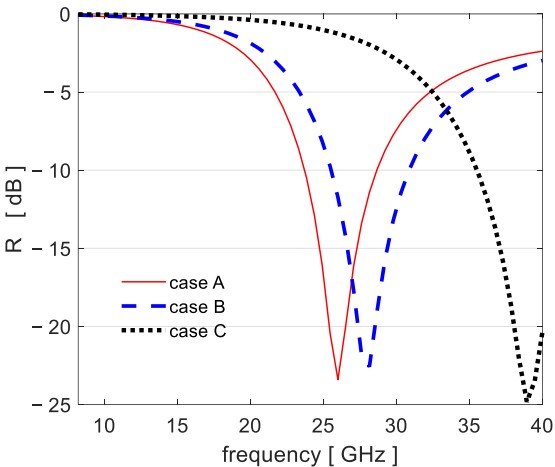

**Figure 10.** Frequency spectra of the reflection coefficient for the cases A,B and C shown in Table 3 for normal incidence.

　　　Finally, the reflection coefficients $R^{TM}$ and $R^{TE}$ are computed for an incidence angle ranging between 0° and 30° and assuming the layer thicknesses reported in Tables 2 and 3 for the composite filled with 2% and 4% of GNPs, respectively. The obtained results are shown in Figures 11 and 12.

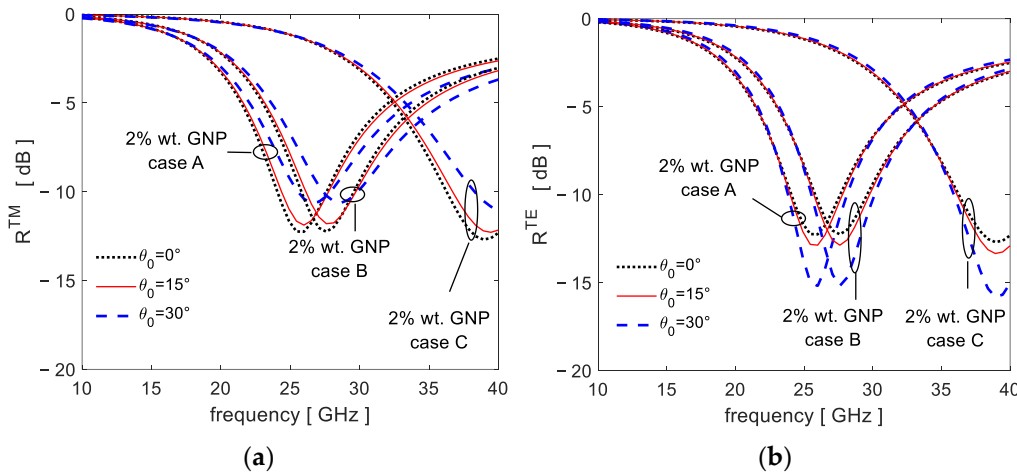

**Figure 11.** Reflection coefficients (**a**) $R^{TM}$ and (**b**) $R^{TE}$ as functions of the incidence angle for the cases A,B and C shown in Table 2 and composite filled with 2% of GNPs.

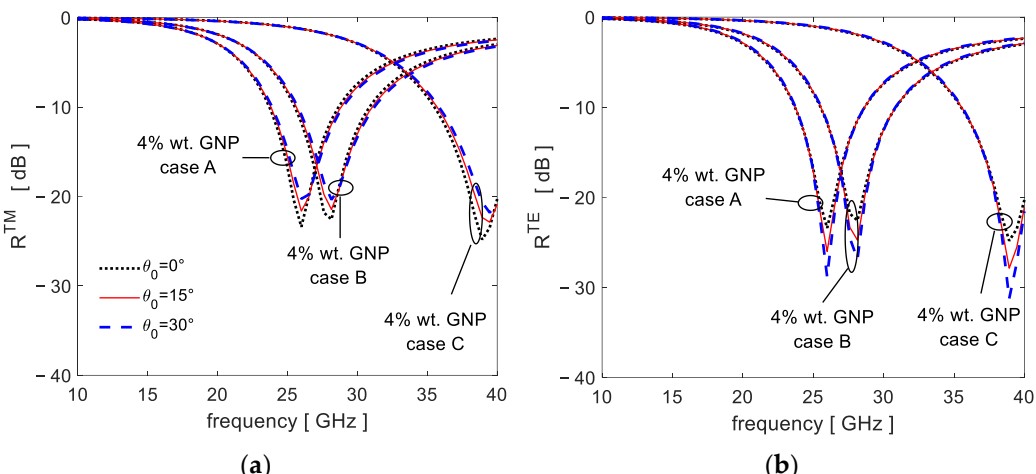

**Figure 12.** Reflection coefficients (**a**) $R^{TM}$ and (**b**) $R^{TE}$ as functions of the incidence angle for the cases A, B and C shown in Table 3 and composite filled with 4% of GNPs.

## 5. Conclusions

New analytical expressions to optimize the thicknesses of two-layers absorbers are derived applying the $(\tanh^{-1})$-function to the hyperbolic matching condition in case of oblique incidence of the impinging plane wave.

It should be noted that the design process only optimizes the spacer thickness if the lossy sheet thickness is selected or vice versa, avoiding time consuming numerical or iterative procedures. On the other hand, absorbing structures designed using complex algorithms, which are able to optimize both the two-layer thicknesses, can be not feasible or of difficult practical realization specially when graphene-based composites, characterized by both dielectric and conductive properties, are selected for the realization of the lossy layer.

The method is tested on absorbers consisting of an epoxy-based vinyl ester resin filled with 2% or 4% wt. of graphene nanoplatelets (GNPs) and a polypropylene dielectric spacer. Both the transverse magnetic (TM) and electric (TE) polarizations of the impinging plane waves are considered. The optimal thicknesses are computed at the target frequencies of 26, 28, and 39 GHz in order to obtain a −10 dB bandwidth able to cover the 5G frequency bands of Europe/UK/China, Korea, and USA between 23.8 and 40 GHz.

It should be noted that, due to the dielectric properties of the composites acting as lossy layers, the theoretical matching condition is not fully satisfied when the optimal thicknesses are considered.

The values of the optimal thickness of the spacer (or composite layer) as function of the selected thickness of the composite layer (or spacer) obtained applying the analytical approach in the case of normal incidence are validated by the comparison with the ones computed numerically considering all the possible combinations of the two-layers thicknesses.

The results obtained demonstrate the effectiveness and high accuracy of the proposed method, making it an ideal candidate for the development of an efficient design tool for absorber design optimization. This tool is of particular significance for this type of graphene based absorbing structure as it allows for the establishment of process parameters for the production of suitable GNP-filled composites, thereby reducing both production efforts and costs.

**Author Contributions:** Conceptualization, M.D. and A.G.D.; methodology, M.D.; software, A.G.D.; validation, M.D. and A.G.D.; formal analysis, M.D.; writing—original draft preparation, M.D.; writing—review and editing, A.G.D. and M.D.; supervision, M.S.S.; funding acquisition, M.S.S. All authors have read and agreed to the published version of the manuscript.

**Funding:** This research received no external funding.

**Data Availability Statement:** Raw data are available upon request.

**Conflicts of Interest:** The authors declare no conflict of interest.

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
