# Peer review of "Optimal Thickness of Double-Layer Graphene-Polymer Absorber for 5G High-Frequency Bands"

_electronics, doi:10.3390/electronics12030588_

Round 1

Reviewer 1 Report

Dear Authors

The manuscript is focused on the new analytical approach to optimize the thicknesses of a two-layer absorbing struc-ture constituted by a graphene-based composite and a polymer dielectric spacer backed by a metallic layer acting as perfect electric conductor (PEC).

The following suggestion and comments should be taken:

1. The authors could insert more numerical data into the Abstract for enhancement of the manuscript.

2. The overall English needs to be improved. Please seek guidance from a native English speaker if possible ("the" "a", commas, plural form and others could be corrected).

3. Please add in the introduction some information about GNPs and their potential applications or modifications. Please cite (1) Appl. Sci. 2020, 10, 1753; doi:10.3390/app10051753  (2) Materials 2020, 13(21), 4975; https://doi.org/10.3390/ma13214975  (3) Polymers 2020, 12(10), 2189; https://doi.org/10.3390/polym12102189

4. Could the authors include the standard deviation of the used methods?

5. Figure 3 please correct this image (description) for better quality.

6. Figure 4b please correct this image (description) for better quality.

7. Why author choose activated carbons for the study? Please explain in the comments.

8. Authors are suggested to describe some future applications in conclusions.

Author Response

please see the attached file

The authors thank the Reviewers for their helpful comments and suggestions. All required changes have been included in the paper and the suggested improvements have been made. The main changes are highlighted in yellow in the resubmitted paper.

Here below, the specific answers to the Reviewers’ comments are reported.

REVIEWER 1

The manuscript is focused on the new analytical approach to optimize the thicknesses of a two-layer absorbing structure constituted by a graphene-based composite and a polymer dielectric spacer backed by a metallic layer acting as perfect electric conductor (PEC).

The following suggestion and comments should be taken:

  1. The authors could insert more numerical data into the Abstract for enhancement of the manuscript.

Authors’ response

The Authors improved the Abstract according to the Reviewer suggestion.

  1. The overall English needs to be improved. Please seek guidance from a native English speaker if possible ("the" "a", commas, plural form and others could be corrected).

Authors’ response

The Authors improved the English with the help of a native English speaker.

  1. Please add in the introduction some information about GNPs and their potential applications or modifications. Please cite (1) Appl. Sci. 2020, 10, 1753; doi:10.3390/app10051753 (2) Materials 2020, 13(21), 4975; https://doi.org/10.3390/ma13214975  (3) Polymers 2020, 12(10), 2189; https://doi.org/10.3390/polym12102189.

Authors’ response

The Authors improved the Introduction providing more information about GNPs and their potential applications or modifications, including also the suggested references.

  1. Could the authors include the standard deviation of the used methods?

Authors’ response

According to the best of Authors’ knowledge, given a set of N () variables Xi, where i goes from 1 to N, the standard deviation  is defined as:

please see the attached file

where  is the mean value of the Xi variables.

The Authors point out that in the submitted paper it is not considered any set of variables. Indeed the Authors compared in Fig. 4(b) the optimal thickness  evaluated numerically with the one obtained by means of the new analytical procedure proposed in this manuscript.

However, for sake of clarity, the Authors computed the relative percent difference between the optimal thickness  evaluated numerically with the one obtained by means of the new analytical procedure.

In particular, the relative percent difference, reported in Fig. 1, is computed as:

Notice that the relative percent difference has been computed only in the decreasing part of the of the d1*(d2) curve reported in Fig. 4(b).

please see the attached file

Fig. 2. Relative percent difference between the optimal thickness d1* evaluated numerically with the one obtained by means of the new analytical procedure.

  1. Figure 3 please correct this image (description) for better quality.

Authors’ response

The Authors improved Figs. 3(a) and (b) and their descriptions.

  1. Figure 4b please correct this image (description) for better quality.

Authors’ response

The Authors improved Figs. 4(a) and (b) and the description of Fig. 4 (b).

  1. Why author choose activated carbons for the study? Please explain in the comments.

Authors’ response

The Authors have choose GNPs filled composites since, as stated in the revised Introduction, <<they allow the production of innovative materials with outstanding electromagnetic properties, suitable for the realization of next generation microwave absorbers>>.

Moreover, <<one of the most investigated absorbing panel is the dielectric Salisbury screen, consisting of a lossy layer followed by a PEC-backed dielectric spacer. However, the main limitation is the thickness that, for frequency applications from a few up to 40 GHz, is in the range of centimeters. In this regard, the use of custom tailored graphene based composites as lossy layers allows to overcome this bottleneck. Therefore, two-layers absorbing structures are attracting ever-growing interest and the optimal design of microwave absorbers consisting of a graphene-based custom tailored composite material and a PEC-backed dielectric spacer is nowadays of primary interest>>.

  1. Authors are suggested to describe some future applications in Conclusions.

Authors’ response

The Authors improved the Conclusions as suggested. In particular, the following sentence has been added: <<The results obtained demonstrate the effectiveness and high accuracy of the proposed method, making it an ideal candidate for the development of an efficient design tool for absorber design optimization. This tool is of particular significance for this type of graphene based absorbing structure as it allows for the establishment of process parameters for the production of suitable GNP-filled composites, thereby reducing both production efforts and costs>>

Reviewer 2 Report

The paper 'Optimal Thickness of Double-Layer Graphene-Polymer Absorber for 5G High-frequency Bands' presents a new analytical approach to optimize the thickness of a two-layer absorbing structure. The performance is tested on absorbers, and the results prove the proposed method's utility and accuracy. The work is valuable and can be published.

Author Response

The Authors are grateful to the Reviewer for her/his positive comment.

Round 2

Reviewer 1 Report

The authors have addressed all comments and the manuscript can be published as is.